# Modelling of the Temperature Difference Sensors to Control the Temperature Distribution in Processor Heat Sink

**DOI:** 10.3390/mi10090556

**Published:** 2019-08-23

**Authors:** Piotr Marek Markowski, Mirosław Gierczak, Andrzej Dziedzic

**Affiliations:** Faculty of Microsystem Electronics and Photonics, Wrocław University of Science and Technology, Janiszewskiego 11/17, 50-372 Wrocław, Poland

**Keywords:** thick film, thermoelectric sensor, thermoelectricity, CPU, numerical modelling

## Abstract

This paper has three main purposes. The first is to investigate whether it is appropriate to use a planar thick-film thermoelectric sensor to monitor the temperature difference in a processor heat sink. The second is to compare the efficiency of two heat sink models. The third is to compare two kinds of sensors, differing in length. The model of the CPU heat sink sensor system was designed for numerical simulations. The relations between the CPU, heat sink, and the thermoelectric sensor were modelled because they are important for increasing the efficiency of fast processors without interfering with their internal structure. The heat sink was mounted on the top of the thermal model of a CPU (9.6 W). The plate fin and pin fin heat sinks were investigated. Two planar thermoelectric sensors were mounted parallel to the heat sink fins. These sensors monitored changes in the temperature difference between the CPU and the upper surface of the heat sink. The system was equipped with a cooling fan. Switching on the fan changed the thermal conditions (free or forced convection). The simulation results showed the temperature gradient appearing along the sensor for different heat sinks and under different thermal conditions. Comparison of the results obtained in the simulations of the CPU heat sink sensor systems proves that changes in the cooling conditions can cause a strong, step change in the response of the thermoelectric sensor. The results suggest that usage of the pin fin heat sink model is a better solution for free convection conditions. In the case of strong forced convection the heat sink type ceases to be significant.

## 1. Introduction

This paper is a part of research leading to developing a method to increase the efficiency of fast processors without interfering with their internal structure.

Processing a large amount of data using the CPU leads to increases in its temperature. To prevent overheating, various techniques are used. The most common method is the combination of a passive heat sink and a fan with adjustable rotation speed [1]. In small devices, such as laptops, active heat sinks are implemented. Active heat sinks transport heat from the processor to the outside of the housing using a heat pipe [2]. Liquid cooling systems or Peltier modules (thermoelectric modules) are also often used [3,4].

The use of integrated temperature sensors allows the use of advanced CPU load control procedures. An example is the scaling of the CPU frequency (dynamic frequency scaling (DFS)) [5] or CPU voltage (dynamic voltage scaling (DVS)) [6] related to CPU temperature. One of the modern approaches is to use not only the knowledge of the CPU temperature, but also knowledge of the thermal conditions prevailing in its surroundings. After all, the cooling efficiency depends on these conditions. If the thermal parameters of the surroundings do not change, it is sufficient to read the temperature measured by a sensor, which is usually integrated inside the CPU. However, if the parameters change, their tracking allows advanced reaction. An example of such a method is TΔT power control [7,8]. It consists of the use of two temperature sensors: an internal CPU sensor (commonly mounted in modern CPUs) and an additional sensor mounted on the heat sink. An additional sensor monitors the cooling conditions of the processor and the ambient temperature. As a result the temperature difference between the CPU and its surroundings is known. 

Any change of the thermal conditions around the processor as well as the cooling heat sink will affect the thermal state of the CPU. However, this will happen with a certain delay, resulting from the heat transport speed through the system. This means the monitoring of the processor's surroundings allows prediction of the conditions in which the processor will be in the future. By knowing external cooling conditions it is possible to react faster to dynamic changes in the environment. Using an appropriate algorithm, it is possible to optimize the CPU's processing, e.g., increasing the number of operations when the ambient and processor temperatures decrease and reducing the number of operations when the temperatures increase. To implement the TΔT method, it is necessary to use the appropriate algorithm and temperature sensors that will provide the required information [7,8,9,10,11,12]. 

Such a CPU load control system requires the presence of temperature sensors that will monitor the processor and heat sink, as well as their surroundings. The presented research concerns a specific part of the project—the design of an appropriate sensor. One of the most important pieces of information will be the actual instantaneous temperature difference between the processor and the environment. Therefore, in this research a thermocouple (thermopile) was used as a differential temperature sensor that monitors the required area. In our previous works [11,12], a hybrid sensor was described that was designed for cooperation with an active heat sink (as in Figure 1a), commonly found in laptops. Such a sensor is not an optimal solution for a classic passive heat sink (example in Figure 1b,c), such as those used in desktop computers.

The designed sensor should monitor the temperature difference between the CPU and its surroundings. This assumption is met by thermocouple sensors. The thermoelectric sensor generates a thermoelectric force when the temperature difference appears between its “hot” and “cold” junctions. In the described system the “hot” junction holds the temperature of the CPU and the “cold” one holds the temperature of the external end of the heat sink. As a result, the precise temperature difference between the CPU and the top of the heat sink fins is measured. If the thermocouples are manufactured as film structures on a flat substrate, they can be mounted in a heat sink, for example in the way shown in Figure 2. In such a system the substrates with sensors act as additional heat sink fins. The arrangement in Figure 2b seems to be a better solution. The “hot” side of the sensor is closer to the CPU (in direct contact) than in the arrangement in Figure 2a. The distant CPU, which is the “cold” side of the sensor, is the same for both cases. The important differences concern the position of the “hot” side. This should be reflected in the level of electrical response of such a sensor, and hence its measurement resolution. The advantage of the arrangement in Figure 2a is easier implementation. During our research, both solutions were compared. Obtained results are shown and discussed later in the paper.

Thermoelectric film sensors on flat substrates can be fabricated using various microelectronic techniques and various materials. Depending on the materials, different resolutions of the sensors can be expected. This is associated with the Seebeck coefficient, which characterizes thermoelectric materials [13,14]. The situation is simplified by the fact that the sensor cooperates with a modern microprocessor system, for which signals with a microvolt amplitude are sufficient for analysis. For this reason, it was decided to use cheap and simple thick-film technology. Many examples of planar, thin-, or thick-film thermoelectric sensors can be found in the literature [15,16,17,18,19]. It was decided to use a ceramic substrate and thermocouples made of cheap materials based on metals. The question did arise of whether a temperature gradient (created along the thermocouple) would be high enough to bring the sensor electrical response to a useful level. For this purpose, it was decided to carry out numerical simulations in the COMSOL Multiphysics 4.3 environment. The simulations were supposed to answer the question of which temperatures gradient will appear for heat sinks with different topologies and in different conditions (free or forced convection).

## 2. Materials and Methods 

### 2.1. The Sensor and the Heat Sink Model

Two models of typical aluminum-based heat sinks were considered for the simulation purposes (please see Figure 1b). The first model consisted of 26 plate fins mounted on a 100 × 100 × 10 mm^3^ square heat sink base (Figure 3a). Each plate fin was 30 mm in height, 100 mm in length, 1 mm in thickness at the bottom, and 0.6 mm in thickness at the top. The second model consisted of 100 pin fins mounted on a 100 × 100 × 10 mm^3^ square heat sink base (Figure 3b). Each pin fin was 30 mm in height, 100 mm in length, with a 1 × 1 mm^2^ cross-section at the bottom, and a 0.6 × 0.6 mm^2^ cross-section at the top. 

The thermal model of the CPU was assumed as the heat source, which was developed for the needs of the project [9,12] with dimensions of 80 × 64 mm^2^ and a heat output of 9.6 W. A cooling fan with a diameter of 100 mm was placed above the heat sink. Depending on the simulation, the fan was turned off (free convection test) or switched on (forced convection test). Two thermoelectric sensors were mounted inside the heat sink (thick-film thermocouples on planar ceramic substrates). They were positioned along the heat sink fins or pins (two of such substrates are visible in Figure 3a,b). The temperature difference along them (in the Z axis) was investigated. The first sensor model, marked “SHORT”, had a length equal to the fin and pin height (30 mm). The second model, marked “LONG”, was 40 mm long (the bottom surface was in contact with the CPU). A small gap was made in heat sink base to insert the LONG sensor (to put it in contact with the CPU).

The temperature difference along the sensor will depend on a number of parameters:Environmental conditions: two variants were considered, namely free convection (fan turned off) or forced convection (fan turned on);Heat output generated by the CPU: 9.6 W was assumed;Dimensions of the planar temperature sensor: two variants of sensors (30 mm or 40 mm long) were investigated (detailed description is provided below);Material of the sensor: the thermoelectric sensor was fabricated on a Low Temperature Cofired Ceramic (LTCC) substrate. A thermopile based on silver and nickel arms was used. A detailed description is given below. Calculations were made to determine the total thermal conductivity of the sensor.

The sensor was designed as a thick-film thermopile consisting of 20 Ag/Ni thermocouples fabricated by screen-printing method on LTCC substrates. Such structures are often used as sensors and energy harvesters [10,11,12,13]. The single thermocouple consists of 2 different materials (in this case Ag and Ni) connected by their ends. If the junctions are held at different temperatures, the thermoelectric force appears in the circuit and the electric current flows through the arms of the thermocouples (the Seebeck effect). If the thermocouple is placed in the area with the temperature gradient, the electrical response of the sensor corresponds to the temperature difference between the thermoelectric junctions. This meets the requirements of the application indicated in this paper. To increase the sensor resolution (level of electric response) the thermocouples can be connected electrically in series and thermally in parallel (see Figure 4c). The electric response of such a device (it is called a thermopile) increases *n* times (*n* being the number of thermocouples). The Seebeck coefficient of the silver-based film is about 1.5 μV/K, and the nickel-based one is about −19.5 μV/K. The total Seebeck coefficient of the thermopile (sensor resolution) is 420 μV/K (20 thermocouples) [13]. The sensor was fabricated using thick-film technology. Ag–Ni thermocouples were screen-printed through a 400 mesh stainless screen using an Aurel VS1520A Stencil Printer. Either alumina or LTCC ceramic was used as the substrate. Subsequently, the structures were fired in a belt furnace at a peak temperature of 700 °C.

Two versions of the sensor were designed, differing in substrate and thermocouple length (please see Figure 2 and Figure 4). The designed LTCC substrate was 200 μm thick, 20 mm wide, and 30 or 40 mm long. These values are typical for structures fabricated with thick-film and LTCC technologies. The width of the designed thermocouple’s arms was 200 μm, the thickness was 100 μm, with 12 μm spacing between adjacent arms (Figure 4). The length of the thermocouples was assumed to be equal to the length of the substrates. As a result, two kinds of sensors were obtained – shorter (called SHORT, with a length of 30 mm) and longer (called LONG, 40 mm long).

### 2.2. The Simulation Parameters

A number of parameters had to be defined to set up the simulations [20,21,22,23]:*γ* = *f*(*ω*, *T*)*N_u_* = *f*(*γ*, *L*, *λ*)Re = *f*(*ρ_a_*, *V*, *L*, *µ*)Pr = *f*(*C_p_*, *µ*, *λ*)Gr = *f*(*g*, *β*, Δ*T*, *L*, *ν_a_*) where *γ* is the heat transfer coefficient (W/m^2^ K); *ω* is the heat flux (W/m^2^); *T* is temperature (K); *N_u_* is the Nusselt number (–); *L* is the characteristic dimension of the heat sink (m); *λ* is thermal conductivity (W/K·m); Re is the Reynolds number (–); *ρ_a_* is the density of the cooling fluid (air) (kg/m^3^); *V* is the velocity of the cooling fluid (air) (m/s); *μ* is the dynamic viscosity of the fluid (air) (kg/m·s); Pr is the Prandtl number (–); *C_p_* is the specific heat of the fluid (air) (J/kg·K); Gr is the Grashof number (–); *g* is acceleration due to Earth’s gravity (m/s^2^); *β* is the coefficient of fluid (air) thermal expansion (1/K); and *ν_a_* is the kinematic viscosity of the fluid (air) (m^2^/s).

Heat transfer by convection and conduction has the largest impact on the analyzed problem (the temperature of the heat sink surfaces is always lower than 70 °C, thus the influence of the thermal radiation will be negligible).

Convection is associated with the macroscopic movement of fluid elements (liquid, gas) [21]. The fluid movement can be forced artificially (forced convection) or can be caused by the buoyancy force resulting from differences in density of certain fluid elements with different temperatures (free convection). The velocity of the flowing fluid (in this case air), its physical properties (viscosity, density), shape, dimensions, and condition of the heat exchanging surface (in this case the heat sink) have a large influence on the value of the convective heat flux [20,21,22,23]. These quantities are expressed through dimensionless variables describing the phenomenon of convection. A dimensionless form of the heat transfer coefficient is the Nusselt number (*N_u_*):(1)$$N_{u}=\frac{\gamma \cdot L}{\lambda }.$$

For the thermal simulations it is necessary to determine the value of *γ*. Therefore, it is necessary to know the Nusselt number for the system. This can be determined using three other variables: Re, Pr, and Gr [21,23].

The Reynolds number (Re) determines the nature of the flow:(2)Re=ρa⋅V⋅Lμ.

The flow can be laminar, turbulent, or indirect between them (transient). It may also not occur at all. The properties of the liquid are included in the Prandtl number:(3)$$\mathrm{Pr}=\frac{C_{p}\cdot \mu }{\lambda }.$$

When dealing with free convection, mass forces associated with gravitational acceleration affect the fluid motion. They are included in the Grashof number:(4)$$\mathrm{Gr}=\frac{g\cdot \beta \cdot \Delta T\cdot L^{3}}{\nu_{a}{}^{2}},$$ where Δ*T* is the temperature difference between the wall and the fluid. In addition to these variables, the geometrical dimension relations included in the *Ki* similarity criteria must be considered. Finally, for steady-state convection, the following dimensionless relationship occurs:(5)Nu=f(Re,Pr,Gr,Ki).

For free convection there is no externally forced flow (Re number) and the Grashof number plays a decisive role. Therefore:(6)$$N_{u}=f\left(\mathrm{Pr},\mathrm{Gr},Ki\right).$$

The course of the phenomenon depends mainly on the layer of fluid adhering to the object's wall (in this case the heat sink), therefore the fluid properties should be determined for the average temperature of the fluid and the wall. For forced convection, the Grashof number becomes irrelevant, whereas the forced flow becomes important. Therefore:(7)Nu=f(Re,Pr,Ki).

Basic parameters were taken in accordance with physical properties tables. The remaining ones were calculated based on previous work [21,23]. Detailed calculations are included in Appendix A. As a result, a heat transfer coefficient for free convection
(8)$$\gamma_{\mathrm{free}}=9\;{W/m}^{2}K$$ and for forced convection (9)$$\gamma_{\mathrm{free}}=54\;{W/m}^{2}K.$$ were obtained. These values were used in simulations.

## 3. Simulation Results and Discussion

A series of simulations were made related to the temperature distribution in the CPU heat sink thermoelectric sensor system. The temperature distributions in free (*γ*_free_ = 9 W/m^2^ K) or forced (*γ*_forced_ = 54 W/m^2^ K) convection conditions were simulated using either the plate fin or pin fin heat sink model. Both the SHORT and the LONG sensors were considered. Temperature distributions were analyzed along these sensors, as well as along the heat sink fins (or pins). The following pictures contain selected simulation results. Figure 5, Figure 6 and Figure 7 show the results for the plate fin heat sink model. A CPU with a thermal power of 9.6 W was assumed as the heat source (located under the heat sink Figure 5). Figure 5a shows a general view of the plate fin CPU heat sink sensor in free convection conditions. The placement of thermoelectric sensors is visible.

Figure 6 shows a cross-section through the plate fin CPU heat sink SHORT sensor system in forced convection conditions. The analogous cross-sections for the LONG sensor as well as for a single heat sink fin were also analyzed (for both free and forced convection conditions). The results of the simulation are summarized in Figure 7, which show the temperature distribution along the CPU heat sink sensor system. The “BLACK”, “BLUE”, and “RED” measurement lines are the temperature distributions along the cross-section lines marked in Figure 6c. Here, Fin_height = 0 marks the CPU heat sink contact. Fin height = 0.01 m is the contact of the solid part of the heat sink with the fin. Fin height = 0.04 m is the upper edge of the fin (or a sensor). For simplicity, the following designations are used:Fin_height = 0 → CPU_HSFin_height = 0.01 m → HS_FIN (or HS_PIN)Fin_height = 0.04 m → TOP.

Figure 7a shows temperature distributions for free convection and Figure 7b shows temperature distributions for forced convection conditions. The Black waveforms represent cross-sections through the solid heat sink fin area, RED through the solid heat sink SHORT sensor area, and BLUE through the LONG sensor area (see Figure 6c). 

In the case of free convection (Figure 7a), the temperature difference between CPU_HS and TOP measured in the solid heat sink fin area (BLACK waveform) is very small (about 0.9 °C).

In the case of a RED waveform, the “cold” edge of the SHORT sensor is in the height HS_FIN. Therefore, the RED and BLACK waveforms are almost identical in the range of CPU_HS ÷ HS_FIN. Over the next few millimeters these temperatures are still very similar. For free convection conditions, the air movement in this volume is very poor, as the air mixes weakly with the higher, cooler layers. As the distance from HS_FIN increases, the temperature difference along the SHORT sensor increases. Finally, it reaches about 2.9 °C. This value is quite small. The electrical response of the sensor (20 Ag/Ni thermocouples, Seebeck coefficient 420 μV/K) is 1.22 mV. 

The LONG sensor is placed in the gap prepared in the heat sink base. The sensor has direct contact with the CPU and weak contact with the heat sink. This is visible in the BLUE waveform in Figure 7a—it strongly deviates from RED and BLACK in the first area (CPU_HS ÷ HS_FIN). The temperature of the “cold” edge of the LONG sensor is at the same level as for the SHORT sensor. However, the temperature difference between “cold” and “hot” thermocouple junctions is larger, at about 4.1 °C, and the electrical response of the LONG sensor is 1.72 mV.

For forced convection (Figure 7b), the temperature difference between CPU_HS and TOP measured in the solid heat sink fin area (BLACK waveform) is also very small, at approximately 1.0 °C. In these conditions the temperature of the entire heat sink is much lower than for free convection (20 ÷ 22 °C in relation to 54 ÷ 58 °C). It is connected with much better heat dissipation from the nearest environment of the CPU heat sink sensor due to strong air movement (fan on). It should be noted that for the purpose for which the sensor was developed, the temperature itself is not very important. Internal temperature sensors integrated with modern CPUs are sufficient for its measurement. The task of the designed sensor is to monitor the changes of the temperature difference between CPU_HS and TOP areas.

The initial part of the RED waveform is again almost identical to the BLACK one, for reasons explained above. Finally, the temperature difference along the SHORT sensor is approximately 1.3 °C. This is a small value, less than half than in the case of free convection. Switching on the fan results in increased air movement, and thus the equalization of its temperature in the volumes between the heat sink fins. As a result, the temperature difference decreases. The electrical response of the SHORT sensor is expected to be 0.55 mV.

Also, for the LONG sensor the temperature difference is much smaller than for free convection conditions (about 2.2 °C, Figure 7b, BLUE waveform). This determines the electrical response of the thermoelectric sensor at the level of 0.94 mV. 

The change from free convection to forced convection conditions causes the mixing of air masses in the close surrounding of the CPU heat sink sensor system. The temperature difference along the sensor is reduced as a consequence. It can be concluded that the stronger forced convection (higher γ_forced_, see Equations (8) and (9)), the smaller temperature difference along the sensor, resulting in smaller voltage in its output. At this point, it should be considered how this affects the possibility of using the thermoelectric sensor to monitor temperature changes in the CPU's environment. The simulation results show that the better cooling of the processor, the smaller the sensor output signal. The question is whether in this case the signal will be useful for the proposed purpose.

To clarify this, it is necessary to specify again that the task is not to monitor the temperature difference between CPU_HS and TOP areas, but to detect quick, instantaneous changes of this difference. If thermal conditions change near the CPU (e.g., forced air movement appears), the sensor's task is to detect this change in a short period of time. The results presented above show the thermal steady state of the system, which will be achieved not immediately but after a relatively long time. 

In the authors opinion, the obtained results prove that when the cooling conditions change (e.g., change from free convection to forced), a high temporary gradient of the temperature difference between CPU_HS and TOP areas will appear. The temperature of the TOP point will quickly decrease from around 57 ÷ 58 °C (Figure 7a) to 21 ÷ 22 °C (Figure 7b). In the surrounding area of the heat sink base the air flow is limited, which causes it mix slower. As a result, the instantaneous temperature difference could be high, which would translate into a high instantaneous response from the thermoelectric sensor (up to a dozen mV). In order to verify this thesis practical tests were carried out, which are described later in this paper.

The results of the simulations should also answer the question of whether the type of heat sink (plate fins or pin fins) has a significant influence on the temperature distribution. To answer this question the simulation results for the pin fin heat sink models are discussed below.

Figure 5, Figure 8, and Figure 9 show the results for the pin fin heat sink model. Figure 5b shows a general view of the pin fin CPU heat sink sensor in forced convection conditions. Placement of thermoelectric sensors is visible.

Figure 8 shows a cross-section through the pin fin CPU _heat sink SHORT sensor system in free convection conditions. The analogous cross-sections for the LONG sensor as well as for single the heat sink pin were also analyzed (for both free and forced convection conditions). The results of the simulation are summarized in Figure 9, which shows the temperature distribution along the CPU heat sink sensor system. The BLACK, BLUE, and RED measurement lines are the temperature distributions along the cross-section lines marked in Figure 6c. Analogous designations were used for the graphs related to the plate fin heat sink model.

Figure 9a shows temperature distributions for free convection, whereas Figure 9b shows temperature distributions for forced convection. The BLACK waveform represents cross-sections through the solid heat sink pin area, RED is through the solid heat sink SHORT sensor area, and BLUE is through the LONG sensor area (please see Figure 6c).

For free convection (Figure 7a) the temperature difference between CPU_HS and TOP measured in the solid heat sink fin area (BLACK waveform) is approximately 1.8 °C. This is significantly more than for the plate fin heat sink model. The volume of the single pin fin is noticeably smaller than the volume of the single plate fin. Thus, the amount of heat reaching the TOP area is reduced. Consequently, the temperature difference is increased, similar to forced convection (Figure 9b). The temperature difference between CPU_HS and TOP measured in the solid heat sink pin area (BLACK waveform) is about 1.6 °C. This is very close to the result for free convection and much more than in the case of the plate fin heat sink model.

In the case of free convection and the SHORT sensor (RED waveform, Figure 9a), the difference is approximately 3.4 °C. For the LONG sensor it is about 4.0 °C (BLUE waveform; Figure 9a). Temperature differences for the sensors are similar to the results obtained for the plate fin heat sink model. The sensors’ volume is the same as for the previous case (the heat transported through them does not change). The shape of the heat sink causes changes in air flow in its surroundings. However, this flow is not too intense for free convection and has a relatively low temperature (43 ÷ 47 °C). For these reasons, the results differ only slightly from those for the plate fin heat sink model.

In the case of forced convection and the SHORT sensor (RED waveform, Figure 9b), the difference is about 1.9 °C. For the LONG sensor it is about 2.8 °C (BLUE waveform, Figure 9b). However, in this case the temperature differences clearly differ from the results obtained for the plate fin heat sink model. The explanation is analogous to that of free convection. The other shape of the heat sink (in relation to the plate fin model) causes changes in the air flow in its surroundings. This affects the cooling conditions, and as a result the temperature difference. This difference is much larger than for free convection, because in this case the air flows intensively (the fan is switched on).

In free convection conditions the temperature of the pin fin heat sink is generally lower than the plate fin heat sink. The temperature of the TOP area is 47 °C in comparison to 58 °C for free convection. The temperature of the bottom of the heat sink (CPU_HS surface) is 42.5 °C compared to 53.5 °C. This is due to the smaller volume of a single pin in relation to a single fin, as described above. The temperature of the heat sink in forced convection conditions is almost identical for both heat sink models. This is a big difference compared to the results achieved for free convection. This suggests that in the absence of forced air flow the usage of a pin fin heat sink is a much better solution. However, for strong forced convection the heat sink model seems to be unimportant.

To verify the correctness of simulation results and the conclusions drawn from them, a verification experiment was carried out. Two LTCC ceramic substrates (DuPont DP951 green tape, thickness 200 μm) were prepared. Their planar dimensions were 20 × 29 mm^2^ and 20 × 39 mm^2^. The Ag/Ni thermocouples were made using the screen-printing method. The DP6145R (Ag) and ESL 2554-N-1 (Ni) pastes were used. The fabricated thermocouple arms had lengths of 28 mm (SHORT substrate) or 38 mm (LONG substrate). The width of both arms was 200 μm, the thickness was about 10 ÷ 15 μm, and the spacing between arms was 100 μm. Both types of sensors consisted of 20 thermocouples. The structures were fired in the time/temperature profile recommended by the manufacturers [24]. Figure 4 and Figure 10 show masks used for screen-printing and photos of ready sensors, respectively. Sensor sensitivity was about 420 μV/K. To summarize, the fabricated sensors were identical to simulated ones.

The sensors were placed inside the plate fin heat sink model (its photos are shown in Figure 11) between its fins. The "hot" edge of the sensors was located near the heat sink base, near the cooled processor (the temperature-controlled heating table was used as the thermal model of the CPU). The "cold" part of the sensors was located near the top of the heat sink—near the edge of its fins, according to Figure 2. Figure 12 shows the sensors mounted in the heat sink. A standard cooling fan was installed above the heat sink. When turned on this operates in forced convection conditions, and when turned off in free convection.

The sensors were mounted as shown in Figure 2. For the SHORT sensor the “hot” side was in contact with the heat sink base, while for the LONG sensor the “hot” side was in direct contact with the heater (in the gap made in the heat sink base). The Keysight 34970A Data Acquisition/Switch Unit was used for the measurements. A pyrometer for non-contact temperature measurement of the heater surface was also used. After switching on, the measuring unit collected the response of the tested sensors and the temperature of the heater in 5-second intervals. The influence of two factors on the sensor output signal was studied:-switching on/off or changing the speed of the cooling fan;-switching on/off or changing the power of the heater (CPU model).

Figure 12 shows the reaction of sensors (voltage response) to sudden, instantaneous changes in cooling conditions for the (a) LONG sensor and (b) SHORT sensor. In the moments marked with the letter "F", the fan was turned on for a short time. This resulted in a change of conditions from free convection to forced convection. A clear step change in the sensors’ responses is visible. The output voltage level is increased by several hundred percent. This is a reaction that will be easily detected by the processor management system. After a few seconds the fan was switched off and the sensor’s response returned to the previous level. This behavior confirms the thesis posed after the analysis of simulation results. During the tests different the fan’s supply voltages were used. This influenced the fan rotation speed, and thus the change of forced convection conditions. As a result, peaks marked in Figure 12 have different amplitudes.

For the characteristics shown in Figure 12, there are also other moments (not marked with the letter "F") of the increase in the sensors output voltage. They are related to the second tested factor—switching on/off and changing the power of the heater.

Figure 13 shows the thermal cycles of the heat sink. The moments in which the heat output was increased are clearly seen. Then, the heat sink temperature was increased (non-contact measurements using a pyrometer). After the heating was switched off, the temperature dropped significantly. This can take several hundred seconds, which is related to the large mass (and thus thermal inertia) of the heater. In the case of the CPU the cooling would be much faster.

In the initial seconds the heater was switched on at a setting of 50 °C. In Figure 13, the temperature increase is visible. It stabilizes after some time at a given level (area marked with the number “1”). After 600 s the heater setting was increased to 90 °C (area “2”) and then to 100 °C (area “3”). After 1320 s, the heating was switched off. A drop in the temperature of the table is visible (area “4”). After 1500 s the intense heating started. When 125 °C was reached the heating was switched off and the table started to cool down (area “5”). A similar cycle was started after 2350 s (area “6”, max temperature 135 °C) and 2900 s (area “7”, max temperature 160 °C). Both tested sensors clearly reacted to changes in the heater power. Both the sensor responses as well as the heater cycles (from Figure 13) are shown in Figure 14. The red lines indicate the moments when the heater is switched on or its power is increased. The temperature of the “hot” ends increases, as well as the temperature difference along the thermocouples. The blue lines indicate the moments the heating is switched off. The temperature of the “hot” ends as well as the temperature difference along the thermocouples decrease. The response of the sensors is immediate and clear (changes in the level of the voltage signal are relatively large), which is important in the context of processor power management.

The last purpose of the paper was to determine whether the SHORT sensor is sufficient to monitor cooling conditions, or if the LONG sensor is needed, as expected, changes in the output signal are larger for the LONG sensor, which has direct contact with the heater surface (thermal model of CPU; Figure 14). However, the level of the SHORT sensor output signal is also high, at the level of several millivolts. This is a sufficient level to be detected and analyzed by a modern processor-based circuit. Moreover, from a practical point of view the SHORT sensor is easier to implement because it does not require much modification of the heat sink base. 

## 4. Conclusions

The paper presents the specific application of thick-film thermocouples (thermopiles) as a temperature difference sensor between the heat source (CPU) and cold (top) side of the heat sink. The obtained simulations results of the CPU heat sink systems show that any change in the cooling conditions can cause a strong step change in the response of the thermoelectric sensor installed inside the system. The results of the simulation indicate that in thermal steady state conditions the temperature difference between the “hot” and “cold” edges of the sensor (and thus along the thermocouples) will be small. However, it should be noted that the use of other thermoelectric materials (e.g., mixed metallic thick- and thin-film thermocouples based on constantan–silver [24], with a Seebeck coefficient of about 43 µV/K) will result in a significant increase of the output signal. Replacing the nickel-based arms with constantan (CuNi) ones will result in a sensor sensitivity increase from 420 to 890 µV/K (20 thermocouples). 

The simulation results were verified experimentally using a thick-film heater (thermal model of the CPU) and two sensors (LONG and SHORT) mounted inside the heat sink. The sensors generate a clear voltage signal at their output when a change in thermal power in the monitored area appears. This was observed in both cases when more power is supplied to the “hot” ends (more heat generated by CPU) as well as when the cooling conditions change (switch on of the fan, changes in convection conditions). If the change in thermal conditions is long-lasting, it begins to affect the entire sensor. In first moment it affects only one “cold” side, therefore a step change in the sensor response is observed. However, the system reaches thermal equilibrium with time. The temperature difference between the “hot” and “cold” sides drops and the voltage signal at the sensor output decreases. After some time, this stabilizes at a low level (as simulation results show). However, if there is another change in the heating/cooling conditions, the next clear step response of the sensor is observed. This is a very beneficial situation due to the planned use of the system in monitoring of thermal changes in the CPU environment to manage its temperature.

The simulations included two models of heat sinks and two kinds of sensors. The results obtained show that in the forced convection conditions, the type of heat sink is of little importance. A slightly lower temperature in the CPU heat sink contact area was obtained with plate fins. The contact temperature was 21.4 °C for the plate fin heat sink and 21.8 °C for the pin fin heat sink, at a CPU thermal power of 9.6 W. This means that in the forced convection conditions the focus should be on providing a high airflow between fins rather than on their shape. However, it is different in free convection conditions. The difference in cooling efficiency between plate fin and pin fin heat sinks is clearly visible in this case. The temperature in the CPU heat sink contact area was 57 °C for plate fins and 46.75 °C for pin fins—the difference is clear. This means that in the pin fin model the air more effectively reduces the heat from the heat sink surface. Similar results were obtained by Micheli et al. [25] when comparing miniature radiators. In this paper, there are 26 fins in the plate fin model with a total base area of 26 cm^2^. This is 26% of the total heat sink base area. For the pin fin model, there are 100 pins with a total base area of 1 cm^2^. This is only 1% of the total heat sink base area. In the forced convection conditions, hot air is removed from the space between fins by a fan. That is why the difference is negligible. In the free convection conditions, the pin fin model leaves more space for gravitational air circulation. This seems to be crucial for the temperature of the system.

The simulation results show that the temperature difference along the sensor is larger in the case of free convection than in the case of forced convection. However, at the moment the fan is switched on or off, a significant, instantaneous change in the temperature difference is observed (thus changing the cooling conditions).

The simulation results suggest that the LONG sensor (with “hot” end in direct contact with the CPU) is a better solution than the SHORT one. The temperature difference along it (and therefore the voltage response of the sensor) is up to several dozen percent higher than in the case of the SHORT sensor. However, LONG sensors require greater modification of the heat sink base, which makes them more troublesome to use. The results prove that the SHORT sensor is an adequate solution (voltage response of the sensor is at a satisfactory level). The results of experimental verification confirm this. The output voltage of the SHORT sensor changed from about 1 mV to more than 2 mV when the fan rotation speed changed slightly. When the speed was increased significantly, the output signal increased to about 8 mV. This is about 800% of the initial value. This would be easily detected by a CPU thermal management system.

## Figures and Tables

**Figure 1 micromachines-10-00556-f001:**
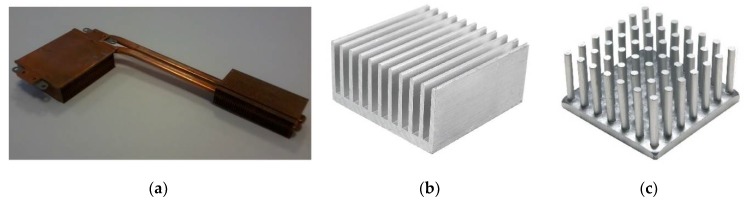
Exemplary commercial CPU heat sinks: (**a**) active heat sink, commonly used in laptops; (**b**) classic passive plate fin heat sink model, typically used in desktop computers; (**c**) classic passive pin fin heat sink model, typically used in desktop computers.

**Figure 2 micromachines-10-00556-f002:**
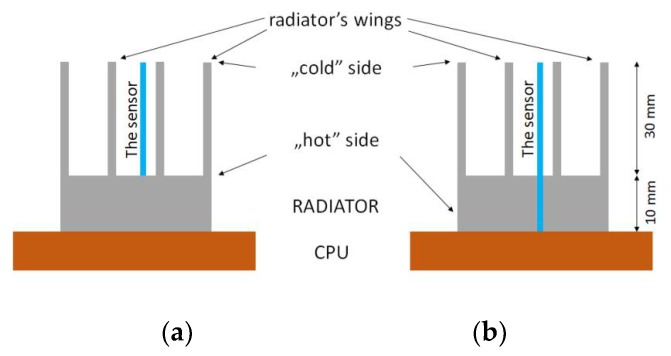
The “SHORT” (**a**) and “LONG” (**b**) sensor models and the ways that they are mounted in the heat sink.

**Figure 3 micromachines-10-00556-f003:**
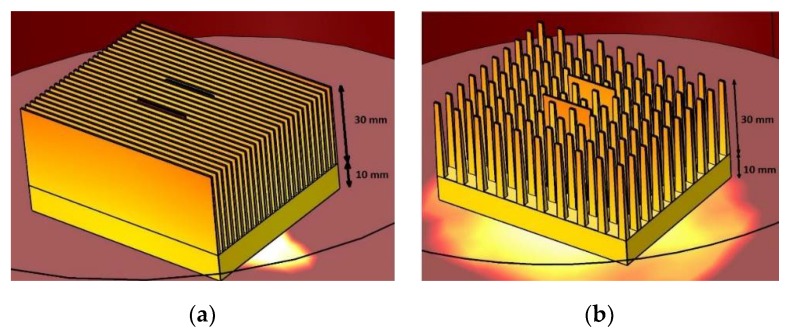
Considered heat sink models: (**a**) plate fins model; (**b**) pin fins model.

**Figure 4 micromachines-10-00556-f004:**
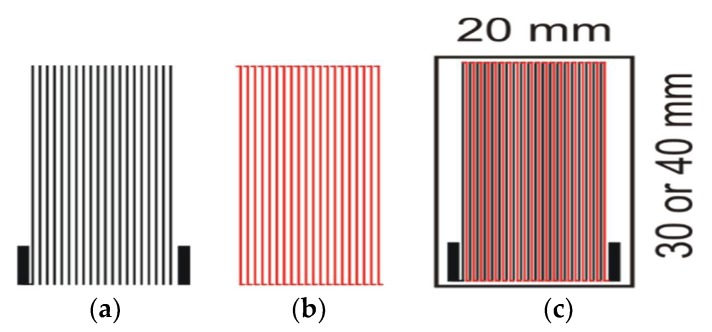
The sensor design: (**a**) mask for screen-printing of Ag arms; (**b**) mask for screen-printing of Ni arms; (**c**) the sensor on the substrate.

**Figure 5 micromachines-10-00556-f005:**
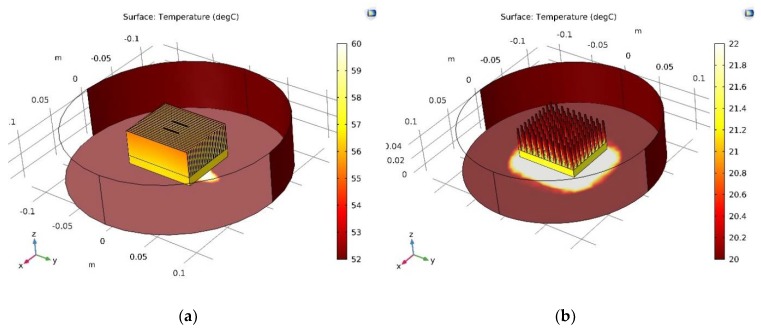
General view of the CPU heat sink sensor system: (**a**) plate fin heat sink model, free convection conditions; (**b**) pin fin heat sink model, forced convection conditions.

**Figure 6 micromachines-10-00556-f006:**
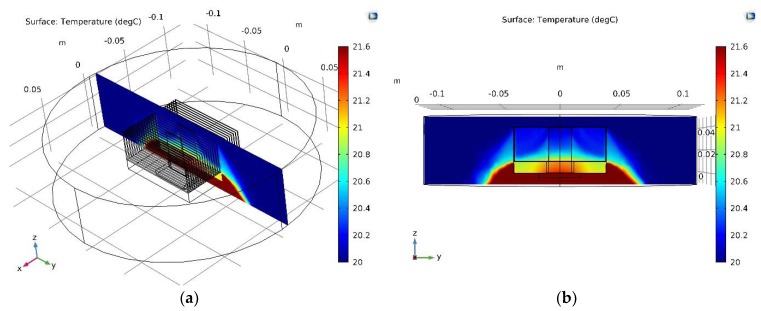
Cross-section through the CPU heat sink sensor volume: (**a**,**b**) temperature distribution on the surface cutting through the heat sink; (**c**) the course of the measurement lines: “RED”, “BLUE”, and “BLACK” (see Figure 7a,b and Figure 9a,b).

**Figure 7 micromachines-10-00556-f007:**
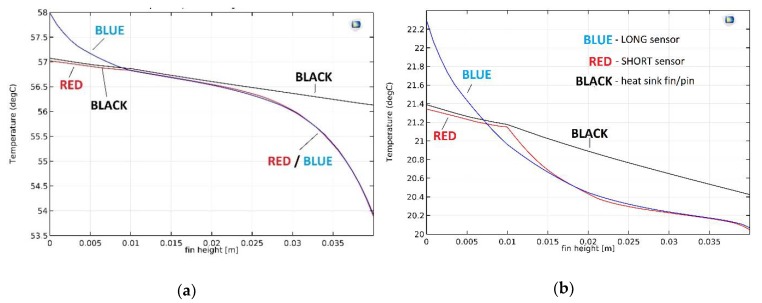
Temperature versus distance from CPU line graphs: (**a**) free convection conditions; (**b**) forced convection conditions. BLACK, RED, BLUE characteristics can be compared with measurement lines in Figure 6c.

**Figure 8 micromachines-10-00556-f008:**
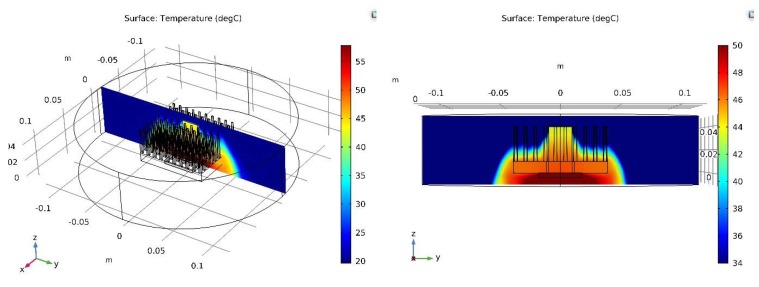
Cross-section of the CPU heat sink sensor volume and temperature distribution on the surface cutting through the heat sink.

**Figure 9 micromachines-10-00556-f009:**
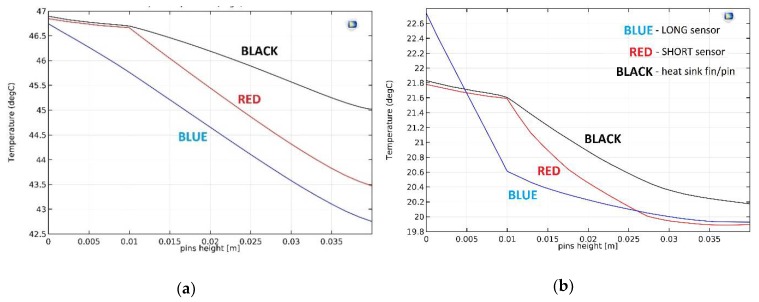
Temperature versus distance from CPU line graphs: (**a**) free convection conditions; (**b**) forced convection conditions. BLACK, RED, and BLUE characteristics can be compared with measurement lines in Figure 6c.

**Figure 10 micromachines-10-00556-f010:**
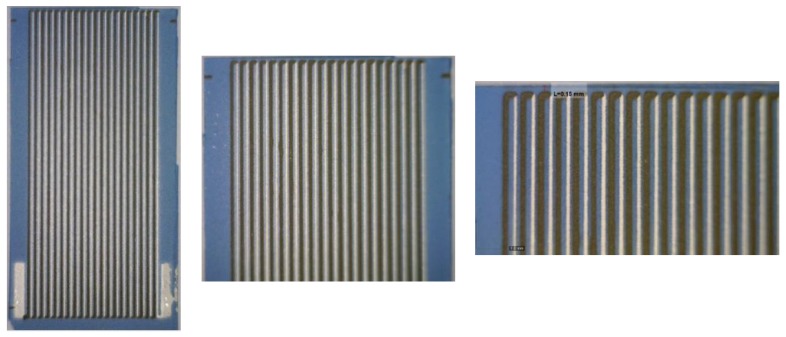
The photos of SHORT structures at various magnifications.

**Figure 11 micromachines-10-00556-f011:**
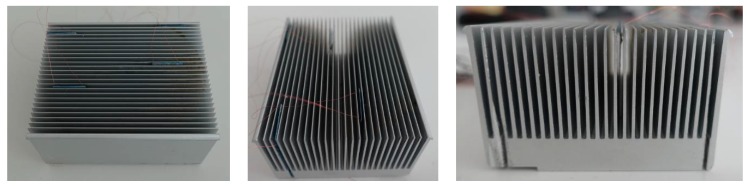
Heat sink with sensors mounted inside.

**Figure 12 micromachines-10-00556-f012:**
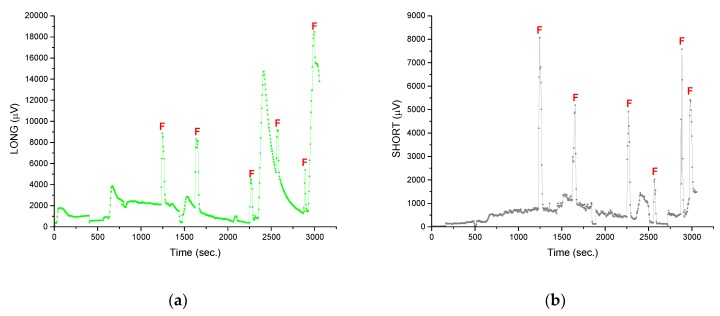
The sensors’ reactions to fan activation (the letter "F" indicates the moments of its activation): (**a**) LONG sensor; (**b**) SHORT sensor.

**Figure 13 micromachines-10-00556-f013:**
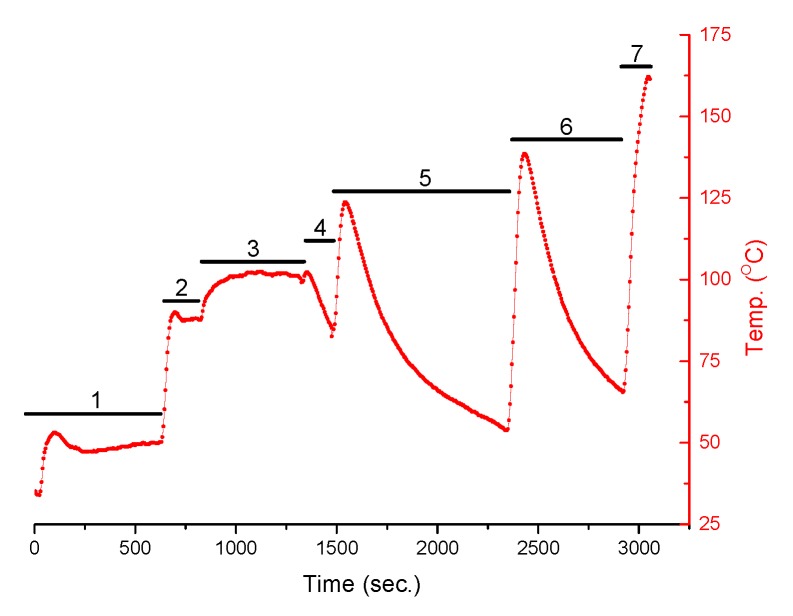
Non-contact measurements of the temperature (thermal cycles) of the heater (thermal model of the processor) using a pyrometer.

**Figure 14 micromachines-10-00556-f014:**
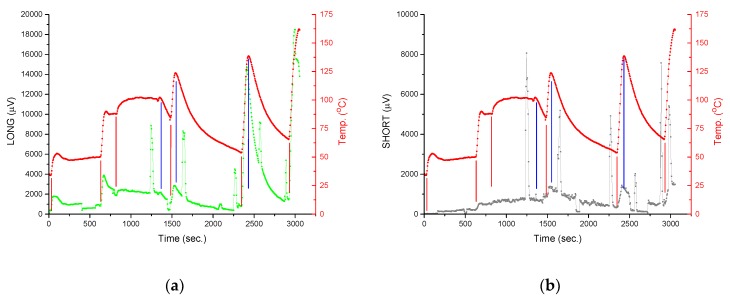
Reactions of sensors to changes in the heater (thermal model of CPU) power: (**a**) green line indicates the LONG sensor response (µV), red line indicates the heater temperature (°C); (**b**) gray line indicates the SHORT sensor response (µV), red line indicates the heater temperature (°C). The moments the fan turned on and off are indicated by vertical red and blue lines, respectively.

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
