# Peer review of "Modelling of the Temperature Difference Sensors to Control the Temperature Distribution in Processor Heat Sink"

_micromachines, 2019, doi:10.3390/mi10090556_

Round 1

Reviewer 1 Report

I read your paper. These are my comments.

Abstract and conclusion should be improved in terms of your three research purposes.

It is not clear to conclude your work based on your three purpose indicated in abstract.

Based on your abstract, " The  third is to compare the efficiency of two types of heat sinks." is not clear with your presentation. Also in conclusion, this fact is not mentioned.

Even you presented the difference between the long and short sensors, it is not clear how its usefulness can be verified.

Author Response

Dear Reviewer,

thank you for your time, the review and suggestions. The paper was revised once again and numerous of language corrections were introduced. I improved the paper according to your suggestions. Many small corrections were made, the most important one is wider discussion regarding to one of the research purpose (comparing the efficiency of two types of heat sinks). The text added to the Conclusion (marked green in the paper):

The simulations included two models of heat sinks and two kinds of sensors. The results obtained show that in the forced convection conditions, the type of heat sink is of little importance. A slightly lower temperature at the CPU/heat_sink contact was obtained for plate-fins. The contact temperature was 21.4°C for plate-fins heat sink and 21.8°C for pin-fins heat sink, at a CPU thermal power of 9.6 W. It means that in the forced convection conditions the focus should be on providing a high airflow between fins rather than on their shape. It is different in the free convection conditions. The difference in cooling efficiency between plate-fins and pin-fins heat sinks is clearly visible in this case. The temperature of the CPU/heat_sink contact was 57°C for plate-fins and 46.75°C for pin-fins. The difference is clear. It means that in the pin-fins model the air more effectively takes off the heat from the heat sink surface. Similar results were obtained by Micheli et al. [29] comparing miniature radiators. In this paper, there are 26 fins in the plate-fins model with a total base area of 26 cm2. This is 26% of total heat sink base area. For the pin-fins model, there are 100 pins with a total base area of 1 cm2. It is only 1% of total heat sink base area. In the forced convection conditions hot air is removed from the space between fins by a fan. That is why the difference is negligible. In the free convection conditions the pin-fins model leaves more space for gravitational air circulation. It seems to be crucial for the temperature of the system.

Reviewer 2 Report

Thanks for addressing the comments from the reviewer. 

Author Response

Dear Reviewer,

thank you for your time and the review. The paper was revised once again and numerous of language corrections were introduced.

This manuscript is a resubmission of an earlier submission. The following is a list of the peer review reports and author responses from that submission.

Round 1

Reviewer 1 Report

I read your manuscript. It is very interesting for heat sink design guide.

But it should be improved or required your response on some comments.

This is on temperature sensor (or thermocouples). Usually, thermoelectric sensor need a contact point. It can be CPU surface.  Then how you need temperature profile along the fin height.

You mentioned temperature difference between hot and cold side. Is it possible to measure both temperature simultaneously by one sensor.

In equation (5), is this equation for forced convection or any general convection?

What does means Ki as a dimensionless number?

What is your sensitivity analysis?

In your contour plot, please do not overlap with legend.(Figs. 6 and 8)

Without the configuration of thermoelectric sensor, your results does not show any feasibility.

In conclusion, you suggested other materials of thermoelectric material.  But it is not related to sensing performance in terms of material.

I recommend to improve your Fig. 6 (c) and others to transfer your idea.  It is not clear what you want to claim your originality.  

Author Response

Dear Reviewer,

thank you for your questions and valuable suggestions. I tried to improve the paper according to them to increase its quality. Answers on a part of your questions you can find below. I hope my explanations will dispel your doubts. I would like to point out that the option „Open peer review: journal will publish the review reports and your responses along with the paper, if it is accepted” was chosen in Micromachines upload system. The following explanations will therefore be visible to the readers of the paper.

Question: “This is on temperature sensor (or thermocouples). Usually, thermoelectric sensor need a contact point. It can be CPU surface.  Then how you need temperature profile along the fin height?”

Answer: Thermoelectric sensor generates the thermoelectric force when the temperature difference appears between its junctions called „hot” and „cold”. In described system the “hot” junction is held in temperature of CPU, „cold” one is held in temperature of external end of the heat sink. As the result the precise temperature difference between CPU and top of pins/fins is measured.

Question: “You mentioned temperature difference between hot and cold side. Is it possible to measure both temperature simultaneously by one sensor?”

Answer: It is not possible by using only the thermocouple. The measurement conception described in the paper base on informations from both, thermopile and the temperature sensor integrated in modern CPUs. In the result both, CPU temperature as well as its change along the heat sink fins/pins is measured.

Question: “In equation (5), is this equation for forced convection or any general convection?”

Answer: It is for any convection. In case of free convection Re becomes negligible (see eq. 6). In case of forced convection Gr can be omitted (see eq. 7).

Question: “What does means Ki as a dimensionless number?”

Answer: Ki describes ratios between geometrical dimensions of investigated object (it influences on thermal convection). In the other equations it appears as eg. C or n (eq. A4) or characteristic dimension L.

Question: “What is your sensitivity analysis?”

Answer: The authors plan to publish another paper, where the practical verification of the results shown will be presented. The design, fabrication, tests and usage of the thermoelectric sensor implemented into heat sink will be precisely described. The sensor consists of 20 silver- and nickel-based arms. The sensitivity of Ag/Ni thermocouple is about 21 µV/K. The sensitivity of thermopile is about 420 µV/K.

Answers on other questions/suggestions you can find in the paper.

Best regards,

Piotr Markowski

Reviewer 2 Report

The objective of this research is not well understood. Authors mentioned that monitoring the thermal state of the processor is necessary to increase the efficiency of fast processors in the first sentences. However, many commercial CPUs can already measure its temperature without the need of additional sensors, as they include an internal temperature sensor. Authors must need to review the current technologies for measuring processors' temperature. Also, authors wanted to integrate a temperature sensor to the processor without interfering with its internal structure. However, as Fig. 7 shows, the ceramic substrate of the "long" sensor acted as an insulator, leading to much higher processor temperature than without the "long" sensor. Thus, the reviewer thinks the developed sensor does not satisfy author's goal. Lastly key information, such as screen printing process, the influence of screen printing on thermoelectric materials, working principle of the sensor, is missing.

Author Response

Dear Reviewer,

thank you for your questions and valuable suggestions. I tried to improve the paper according to them to increase its quality. Answers on a part of your questions you can find below. I hope my explanations will dispel your doubts. I would like to point out that the option „Open peer review: journal will publish the review reports and your responses along with the paper, if it is accepted” was chosen in Micromachines upload system. The following explanations will therefore be visible to the readers of the paper.

Question: “The objective of this research is not well understood.”

The introduction (lines 29-71) was improved to make objectives more clear.

Question: “Authors mentioned that monitoring the thermal state of the processor is necessary to increase the efficiency of fast processors in the first sentences. However, many commercial CPUs can already measure its temperature without the need of additional sensors, as they include an internal temperature sensor. Authors must need to review the current technologies for measuring processors' temperature.”

It is true. Modern CPUs contains integrated temperature sensor. But to implement the TΔT control technique it is necessary to know both, the CPU and the boundary temperature (on the top of fins/pins) – see references [7,8]

Question: “Also, authors wanted to integrate a temperature sensor to the processor without interfering with its internal structure. However, as Fig. 7 shows, the ceramic substrate of the "long" sensor acted as an insulator, leading to much higher processor temperature than without the "long" sensor. Thus, the reviewer thinks the developed sensor does not satisfy author's goal.”

It is true, the thermal conductivity of ceramic substrate is much lower than of Al-based heat sink. However, sensor/CPU contact area is very small in comparison to CPU surface (see Figs. 2, 3, 6): 0.2 mm * 20 mm = 4 mm2. CPU surface: 80 mm * 64 mm = 5120 mm2. The sensor influence on heat transfer is negligible. SHORT version of the sensor was also investigated in the paper. It has no direct contact with CPU. SHORT sensor is resolution is lower than LONG one, but still sufficient to indicated application.

Question: “Lastly key information, such as screen printing process, the influence of screen printing on thermoelectric materials, working principle of the sensor, is missing.”

The authors plan to publish another paper, where the practical verification of the results shown will be presented. That's why this thread has not been developed in the current publication. The Authors use screen-printing method to thick film thermocouples fabrication for long time. Thermoelectric parameters of Ag/Ni thermocouples were characterized and described eg. reference [11,12,19]. Fabricated thermopiles were used as sensors as well as energy harvesters. According to Reviewer suggestions the paragraph related to working principle of the sensor, screen-printing process was added to the paper (lines 122-137)

Answers on other questions/suggestions you can find in the paper.

Best regards,

Piotr Markowski

Round 2

Reviewer 1 Report

I read your revised paper. I can recommend that you should improve your paper with some verification or validation without making another paper. Also your presentation and descriptions of working principle should be updated. 

Reviewer 2 Report

The paper itself should validate the results. The validation of the work must not be future work. Authors mention that experimental validation will follow in the next paper. However, readers and reviewers cannot believe the results, unless the paper proves it. There are different ways to validate the results other than experiments, e.g. comparing simulation result with theoretical model.

Discussion is not concise and clear to understand. Authors must not list all the details of the results, but focuses on explaining key questions. Currently, the discussion section does not clearly present and solve the important questions.

Figures are not clear to understand, e.g. are heat sinks in Fig.1. from authors' work?; Fig. 3 misses color bars; Fig. 4c what is the height of the mask? Is it 30 or 40mm?; There are figures with fonts overlaping.